# Heterogeneity and Temporal Stability of Residential Water Use Responsiveness to Price Change

**Masayoshi Tanishita \*** and **Daisuke Sunaga**

Department of Civil and Environmental Engineering, Faculty of Science and Engineering, Chuo University, Tokyo 112-8551, Japan; dsunaga.385@g.chuo-u.ac.jp
\* Correspondence: mtanishita.45e@g.chuo-u.ac.jp; Tel.: +81-338-171-810

**Abstract:** Many papers estimate the price elasticity of water demand. However, heterogeneity and temporal variation of price elasticity of residential water use are still unclear. We analyze these issues by applying the latent class analysis and t-test using disaggregated data of approximately 30,000 households recorded over five years: Two years before and three years after a tariff revision. As a result, the households are divided into three (heterogeneous) groups: About 5% of them responded to the price change sensitively, 40–60% slightly, and 35–55% not at all. Households with high water use prior to the revision had higher price elasticity. In addition, the price elasticity in the first and third years after the revision was the same.

**Keywords:** heterogeneity; temporal stability; price elasticity; residential water use; latent class

## 1. Introduction

Water is an indispensable commodity in daily life and necessitates efficient supply and restricted use [1,2]. On the other hand, access to affordable water and sanitation services is crucial for realizing the human rights to water and sanitation [3]. A large number of studies have argued that the commodification of water should retain its "public good status" [4–7].

To ensure its affordability for all and a sustainable water system, we must develop appropriate tariff structures. Certainly, the lower the price, the better—but if it engenders water wastage, it will result in an overwhelming increase in supply costs. In Japan, where the study was conducted, water tariffs have increased to recover the cost for renewing water pipes and water facilities. Recently, water services in Sendai City have been privatized for the first time in Japan to reduce the burden on users, due to population decline and aging equipment.

In many countries, including Japan, tariff consists of a basic charge and an increasing block tariff. In addition to efficiency, another important issue of a tariff structure is equity. Price elasticity is a useful measure incorporated in this discussion.

Many empirical studies have been conducted to date; a meta-analysis has been conducted to collect and analyze the estimates of price elasticities [8–13]. The average price elasticity of water demand was −0.51 to −0.36. Few studies have found that household water demand is price elastic [14,15]. Functional specification, aggregation level, data characteristics, and estimation issues are associated with different elasticity values. Spatial variations in price elasticities have also been documented. Dalhuisen et al. (2003) reported that price elasticities are lower in Europe than in the United States, and price elasticities within the United States are greater in the arid West regarding the absolute value [10].

Thus, although it is known that the absolute value of elasticity is less than one, few studies have focused on heterogeneity and temporal variation. What are the attributes of households that respond significantly to price changes? How many households do not (or cannot) respond to price changes? In general, water supply agencies change the tariff moderately. If the price change is small, some households may not change their water use. In addition, Wathieu (2004) proposed a mechanism of consumer habituation. In his

proposal, demand is dependent on the price level and initial habit conditions, and interpreting consumer price-sensitivity based on temporary price changes is misleading [16]. Do households that reduced their water use immediately after the price revision continue to do so after some time has passed? In other words, is the response to a price change temporary or not? These are major questions of this paper.

When we use aggregate data, it is difficult to analyze differences in household attributes. Therefore, many researchers treat household panel data. They suggest lower-income groups were more price-responsive than higher-income groups. For example, in Cyprus, Hajispyrou et al. (2002) reported a price elasticity of $-0.79$ for the lowest-income group, as compared to $-0.39$ for the highest-income group [17]. In Belgium, the price elasticity for the lowest-income quintile was estimated as $-0.76$, as compared to $-0.25$ for the highest-income quintile [18]. Mayol (2017) showed that by observing the different types of consumers, some consumers were more sensitive to windfall effects caused by a lower tariff, but that most consumers were incentivized to reduce their consumption [19]. Reynaud et al. (2018) showed that households in single-family units are more reactive to changes in water price than households in multi-family units, all being equal [20]. On the other hand, Pashardes et al. (2002) demonstrated the price elasticity estimation for each income group and clarified that price elasticity increased with an increase in income [21]. Brolinson (2020) used billing records and demographic data to indicate that wealthier households were more price elastic than lower-income households [22]. Recently, EL-Khattabi et al. (2021) found that heavy-usage households were more price-sensitive than other households, and price elasticity was largely invariant to household wealth [23]. Taken together, these previous results suggest that we still have room for discussion regarding heterogeneity.

In addition, Zikos (2008) showed that a system based on information, normative motivation, and regulation solved water shortage problems better than systems based on pricing [24]. Between 1989 and 1993, water use fell as the result of public awareness campaigns and regulations. Since then, use has increased substantially despite the introduction of price incentives. He emphasized those two mechanisms seemed to be involved. First, many people are not aware of the level of the water price, implying that the price was not an important factor when deciding upon water use. Second, he argued that moral persuasion might be more effective. Using prices weakens the effect of allowing use norm. These suggest the price elasticity is zero for some households.

Regarding the temporal variation, in the analysis using aggregated time-series data, short- and long-run elasticities were obtained using the lag term in the specification. Short-run elasticities are often lower than their long-run counterparts. This suggests that consumers may require time to adjust to water-using capital stocks and study the effects of use on their bills. However, this relationship is derived from the model specification. And in some papers using panel data, the authors discuss the seasonal variance, but a few papers discuss the temporal stability of the responsiveness to price changes.

Furthermore, to the best of our knowledge, analysis using disaggregated data has not been conducted in Japan. In addition, there is insufficient research on the difference in elasticity depending on the consumer type and the change in elasticity after the revision.

Another issue regarding the response to water price change is the definition of unit price. Standard economic theory predicts consumers will optimize at the margin, to the point where marginal benefit equals marginal cost. However, Ito (2014) showed strong evidence that consumers respond to average price rather than marginal or expected marginal price [25]. One explanation is that utility bills are often complex and make it harder for consumers to understand the nonlinear structure of their pricing. Then the average price is applied for estimation [14,23,24]. Recently, Mayol and Porcher (2019) also showed consumers did not always react to marginal prices [26]. We discuss which definition is more suitable in Japan.

In this paper, we estimated price elasticity using monthly data from approximately 30,000 households. The data were collected for a total of five years: Two years before

the tariff revision, and three years after the revision. Simultaneously, we presented the relationship between price elasticity and water use using latent class analysis. We also compared the elasticities between the first and third years after the revision. In addition, we compared the marginal cost and the average cost in terms of unit price definition to calculate the price elasticity.

## 2. Materials and Methods

The data for this analysis were obtained from households in Hadano City, Kanagawa Prefecture, Japan. Hadano city has about 160 thousand population, and this is in the top 10% of Japanese municipalities. The average household size is 2.4, and the share of single-person households is 34% in 2017. Water service is operated by the municipality. Water use is measured every two months, and payment is made in the following month.

The tariff revision was performed in April 2016 (Table 1). The basic charge was increased by about 31%, and the metered charge was also increased by about 2–21%. There was no special event, such as heatwaves, awareness campaign, and income change, etc., during this period except for the price change. The data include bi-monthly water use and a fee for each household from 2014 to 2018. That is, two years before the revision and three years after the revision. Combining with Table 1, we can obtain (a) average price by dividing the fee (Yen) by the water use ($m^3$) and (b) marginal price by reading water use. Unfortunately, information was not available on the household income or the number of households.

**Table 1.** Tariff structure.

| Tariff Revision | Basic Charge | Metered Charge ($m^3$) | | | | | |
|---|---|---|---|---|---|---|---|
| April 2016 | −8 | 9–20 | 21–30 | 31–50 | 51–100 | 101–500 | 501~ |
| Before (Yen) | 520 | 70 | 80 | 130 | 195 | 220 | 220 |
| After (Yen) | 680 | 85 | 95 | 140 | 205 | 225 | 245 |

In Hadano city, almost half of the households have meter readings during even-numbered months, while the rest have meter readings during odd-numbered months. Households in these two groups are completely different. In other words, there are two datasets.

We selected households that paid a fee continuously for five years and used 16 $m^3$ of water for two months. Households that experienced more than a twofold change in average annual water use were assumed to have experienced a significant change in the number of household members or income, and were excluded from the analysis. Then, 14,544 households in even-numbered months and 15,519 households in odd-numbered months were considered for this study. The data for May and April, which observed mixed effects before and after the revision, were not used. Monthly average temperatures were also collected over a period of five years.

Descriptive statistics are shown in Table 2. In this study, the residential water use before the tariff change was averaged from the data collected between 2014 and 2015. The values for post-revision were averaged from the data collected between 2016 and 2018. Therefore, we obtained five price elasticities (point estimates) for each household every two months. If we assume that each household has one price elasticity, the variance of these price elasticities would be small.

Price elasticity is the value obtained by dividing the rate of change in water use by the rate of price change. Elasticity has a negative sign when water use decreases, due to rising prices.

**Table 2.** Descriptive statistics.

| | Tariff Revision | Min. | 1st Qu. | Median | Mean | 3rd Qu. | Max. | Std.Dev |
|---|---|---|---|---|---|---|---|---|
| Water use | Before | 16 | 34 | 45.5 | 48.6 | 59.5 | 147.5 | 19.2 |
| (m³/two months) | After | 16 | 34 | 44.7 | 47.9 | 58.7 | 136.7 | 19.0 |
| Marginal price [Yen/m³] | Before | 70 | 70 | 80.0 | 89.4 | 80 | 195 | 25.6 |
| (Price1) | After | 85 | 85 | 95.0 | 103.2 | 95 | 205 | 25.6 |
| Average Price [Yen/m³] | Before | 70.1 | 73.1 | 75.0 | 78.2 | 78.8 | 137.5 | 8.5 |
| (Price 2) | After | 69.0 | 91.8 | 93.1 | 96.0 | 96.3 | 145.5 | 7.0 |
| Average Temperature [°C] | Before | 4.7 | 9.4 | 16.1 | 15.7 | 22.3 | 26.8 | 7.6 |
| | After | 3.9 | 8.2 | 16.8 | 16.0 | 22.3 | 27.9 | 7.8 |

We assume that each household has an elasticity value, and the value is the same before and after the price revision. We also assume that the observed water uses are affected by temperature and included errors. We applied latent class analysis if the price elasticities of households followed a mixed normal distribution and adjusted the average temperature difference [27,28], and used the flexmix package in the CRAN R for estimation [29].

$$H(y|w, \Theta) = \sum_{s=1}^{S} \pi_s(w, \alpha) N\left(y \middle| \mu_s, \sigma_s^2\right) \tag{1}$$

$$\pi_s(w, \alpha) = \frac{exp(w\prime\alpha)}{\sum_{s=1}^{S} exp(w\prime\alpha)} \tag{2}$$

where $S$ is the number of classes, $N\left(y \middle| \mu_s, \sigma_s^2\right)$ is the Gaussian distribution with mean $\mu_s$ and variance $\sigma_s^2$. $\Theta$ denotes the vector of all parameters of the mixture distribution, and the dependent variables (elasticities) are y. The concomitant $w$ and $\alpha$ are parameters. These parameters are estimated using EM algorithm.

Furthermore, we analyzed the relationship between price elasticity and water usage. Subsequently, a significant difference in elasticity was tested for 2016 and 2018, which were the first and third years after the revision, respectively.

Figure 1 shows a scatter plot of residential water use before revision and price change rate. The average tariff change rate was approximately 20%, but the rate of change in tariffs was lower for households that used more water.

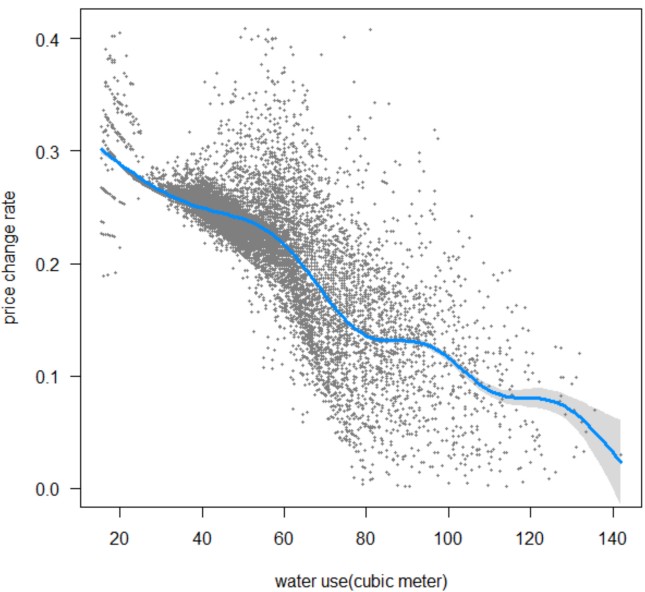

**Figure 1.** Scatterplot of water use and price change rate (Price 2).

## 3. Results

### *3.1. Relationship between Water Use and Elasticity*

Figure 2 shows a scatter plot of the water use before tariff revision and elasticity in June. Notably, many households had a positive elasticity. That is, the water use increased after the rate revision. This does not imply that residential water use has increased, due to the price revision, but it should be interpreted as the change in residential water use by the tariff revision being within a certain fluctuation range, and the households with increased water use were unresponsive to price variations.

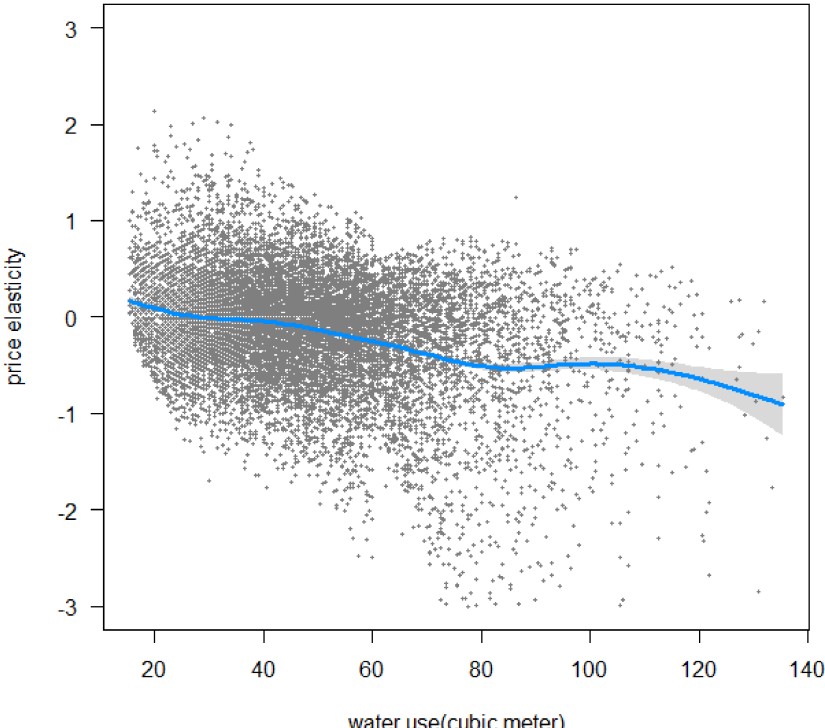

**Figure 2.** Residential water use before the tariff revision versus price elasticity in June. The shade shows 95% significant intervals.

Next, observing the relationship of water use prior to the revision, the higher the water use in any month, the greater the elasticity. As shown in Figure 1, the rate of change in unit price decreased for households that used more water, and such households had relatively larger elasticity. Conversely, it was difficult for households with relatively less water usage to reduce it further, even with a price change.

The variation was relatively large for Price 1 (marginal price) as compared to Price 2 (average price), and had a higher proportion of households with an absolute value exceeding 1 (Figure 3).

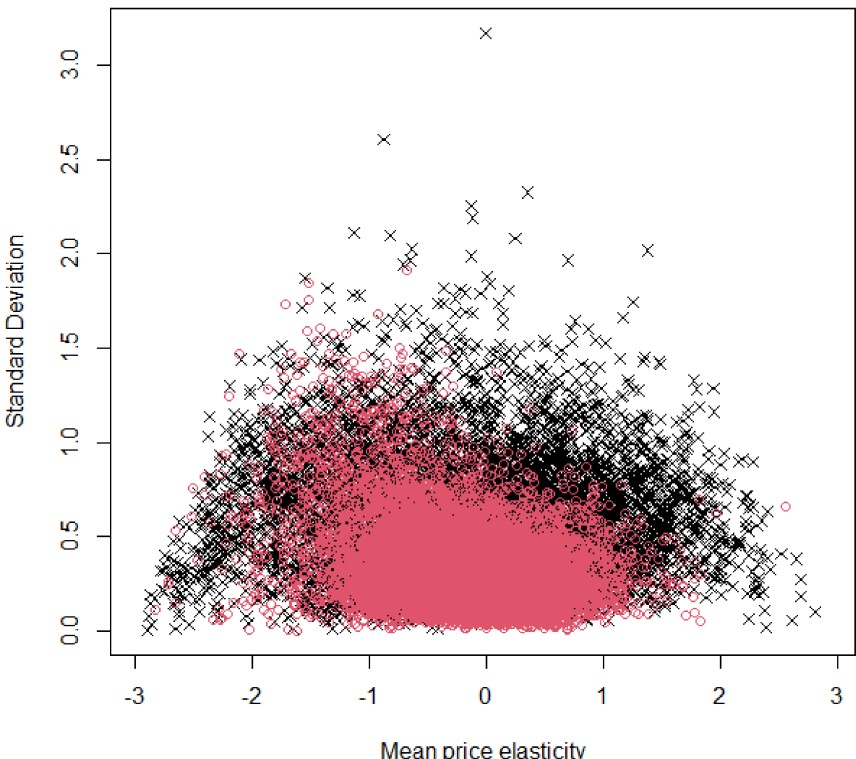

**Figure 3.** Mean versus standard deviation of price elasticity of each household in. odd-numbered months (Black cross: Price 1 (marginal price), Red circle: Price 2 (average price)).

### 3.2. Latent Class Analysis

Regression analysis was performed by treating each household as a random effect, using the difference in average temperature before and after the tariff revision as an explanatory variable. Next, the elasticity value was adjusted, assuming that there was no visible temperature difference. We subsequently applied the flexmix package in the latent class analysis for even and odd months.

In both even- and odd-numbered months, three classes were selected when the Bayesian information criterion (BIC) was used for the index (Figure 4). The results are shown in Table 3. A class with high elasticity responded significantly to price revisions, a class with low elasticity responded slightly to the tariff revision, and a class with zero elasticity remained unresponsive to tariff revision. In addition, it was estimated that 35–55% of households were unresponsive to the revision, and 5–6% of households responded significantly to price variations.

The weighted average of price elasticity was −0.10 and −0.11 in even-numbered and odd-numbered months, respectively. Interestingly, the households considered in both months were entirely different, but the estimated price elasticity was almost similar.

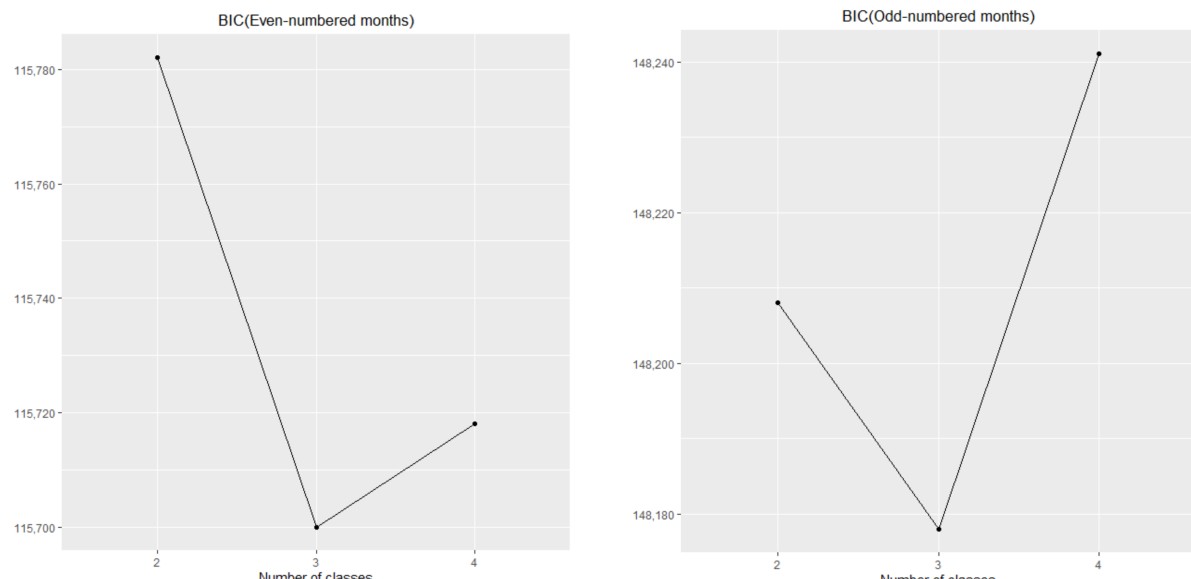

**Figure 4.** Relationship between BIC and number of classes.

**Table 3.** Estimated results of latent class analysis.

| Even-Numbered Months | | | | Odd-Numbered Months | | | |
|---|---|---|---|---|---|---|---|
| **Class ID** | **2** | **1** | **0** | **Class ID** | **2** | **1** | **0** |
| share | 0.06 | 0.59 | 0.35 | share | 0.05 | 0.41 | 0.55 |
| mean | −0.55 | −0.09 | 0.00 | mean | −0.61 | −0.16 | 0.00 |
| std. error | 0.05 | 0.01 | 0.01 | std. error | 0.03 | 0.01 | 0.01 |
| weighted mean | | −0.10 | | weighted mean | | −0.11 | |
| Number of samples | | 69,027 | | number of samples | | 57,318 | |
| BIC | | 115,700 | | BIC | | 148,175 | |
| BIC for lm | | 122,283 | | BIC for lm | | 151,989 | |

*3.3. Difference in Elasticity between the First Year and the Third Year after Revision*

Finally, the elasticity values were compared after similarly adjusting the temperature difference for the first and third years post-revision.

The results are shown in Figure 5. There was no significant difference in price elasticity between even-numbered and odd-numbered months. The effect of tariff revision was not confirmed for the three years.

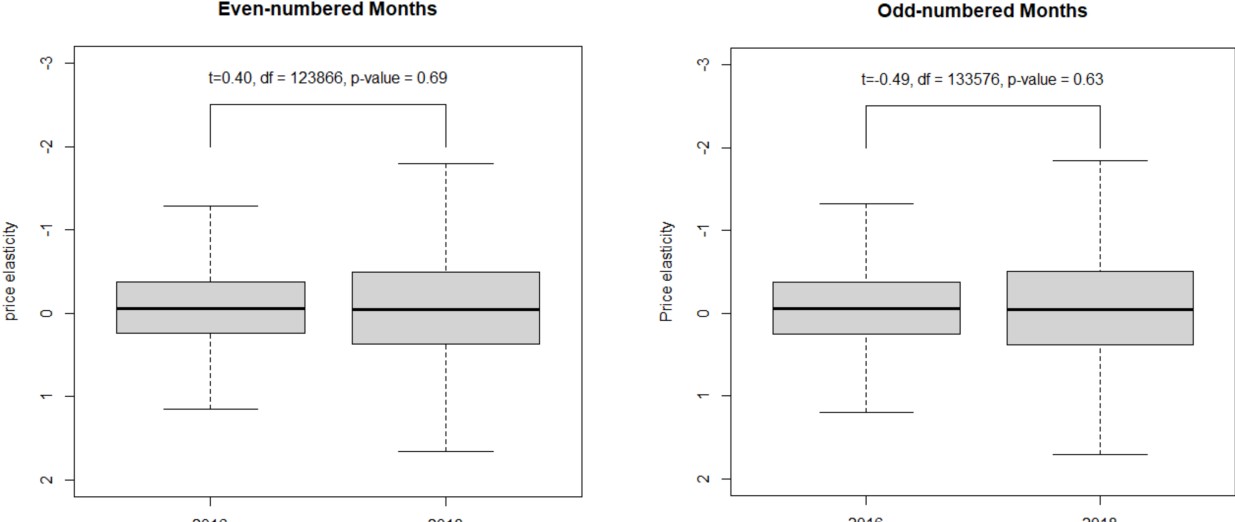

**Figure 5.** Comparison of price elasticity between the first year (2016) and the third year (2018) after the revision. Note: t in the figure shows the t-value.

## 4. Discussion

In this study, we analyzed the heterogeneity and temporal stability of price elasticity of residential water use using disaggregated panel data before and after price revision. We demonstrated that the price elasticity increased with increased water use. It was estimated that approximately 5% of households responded significantly to the price change, 40–60% of households responded slightly, and 35–55% were unresponsive to the changes. The estimated mean price elasticity ranged between −0.11 and −0.10. Furthermore, the difference in elasticity between the first and third years after the revision was insignificant.

What we found that 35–55% of households were unresponsive to the price change. There are two possibilities for this explanation. One explanation is that households do not recognize or care about the price and its change. In Japan, the ratio of water bill to total expenditure is about 1.7%. Compared with food and communication cost, the water bill is minor for households. The other explanation is that it is difficult for households with relatively less water usage to reduce it further. As shown in Figure 1, the price change rate is higher for light-usage households. Therefore, light-usage households can easily recognize the price change. However, their elasticity was so small.

In general, households that use more water have higher incomes and a larger household size. The price elasticity was estimated to be higher for such households. As observed by Brolinson (2020) and El-Khattabi et al. (2021), heavy-usage households are more price-sensitive than other households, and households with increased economic margins were more likely to adjust their use [21,22]. In the future, when increasing the tariff to cope with the increasing maintenance cost, it may be desirable to raise the price rate per unit rather than charging a flat rate.

Standard economic theory predicts consumers will optimize at the margin, to the point where marginal benefit equals marginal cost. We compared the marginal cost and the average cost in terms of unit price definition. The variation of price elasticity among households for Price 2 (average price) was relatively smaller than that for Price 1 (marginal price), and a lower proportion of households with an absolute value exceeding 1 using Price 2. Therefore, we concluded that average price is more appropriate for the unit price definition. In other words, consumers are less aware of the marginal price. If consumers scan their bills and notice higher water charges, the average price is likely to be more reasonable. Sallee (2014) showed the rational inattention, due to the high information cost of knowing both one's electricity usage, as well as the price faced in a nonlinear tariff [30]. Shaffer (2020) showed some consumers mistook the marginal price to be the average price,

rather than responding to the average price in lieu of the marginal price [31]. Furthermore, Mayol and Staropoli showed that subjects preferred simple tariffs over complex ones. However, when they receive adequate information about tariffs and appropriate behaviors, they choose more complex tariffs. These results argue in favor of self-selection of tariff forms, to account for consumers' different abilities to respond to the price signal [32]. We need further discussion to understand the consumers' behavior. However, if households do not react to price, nonlinear pricing would be unsuccessful in achieving its policy goal of water conservation and critically change the welfare implications of nonlinear pricing.

Finally, given the low water price elasticity, pricing schemes may not be an effective tool to change household water behavior. Non-pricing measures, such as water conservation programs, education campaigns, or smart metering, and so-called nudge measures, would be alternatives.

In Europe, the privatization of municipal public service provision has been a phenomenon since the 1980s, but in the last decade, there is increasing evidence of trends in the opposite direction. Contract reversals to direct public management are increasingly frequent. In Chile, water privatization in 1981 brought high concentration and inequality in the distribution of water rights [5]. Privatization, which has started in Japan, may not be the solution for rising costs for equipment renewal and improving management efficiency in response to population decline. Participatory budgeting is increasingly common in municipalities worldwide, enabling residents to agree on the budget priorities of local government [33]. Such processes have often contributed to improving basic service provision, including water and sanitation.

Unfortunately, data attributed to households, such as income and household size, could not be obtained. Therefore, this elasticity may have omitted the variable bias [34–37]. In addition, analyzing the data of odd-numbered months and even-numbered months in an integrated manner is a future task. Furthermore, the optimal tariff structure, including the impact on business water demand, should be discussed in the next step [1,38–40].

**Author Contributions:** Conceptualization, M.T. and D.S.; Methodology, M.T.; Software, M.T.; Validation, M.T. and D.S.; Formal Analysis, M.T.; Investigation, M.T.; Resources, M.T.; Data Curation, M.T.; Writing—Original Draft Preparation, M.T.; Writing—Review and Editing, and D.S.; Visualization, M.T.; Supervision, M.T.; Project Administration, M.T. All authors have read and agreed to the published version of the manuscript.

**Funding:** This research received no external funding.

**Institutional Review Board Statement:** Not applicable.

**Informed Consent Statement:** Not applicable.

**Data Availability Statement:** We submitted our data to Dryad. Tanishita, Masayoshi (2021), Water demand, price and temperature in 2014-18 in Hadano City, Kanagawa, Japan, Dryad at https://doi.org/10.5061/dryad.pnvx0k6mk (accessed on 25 August 2021), ID, year, month, water use ($m^3$, "water"), marginal and average price (Yen/$m^3$, "price1" and "price2", respectively) and temperature (Celsius, "tenp") in each line.

**Acknowledgments:** We thank Takashi Shimura at Hadano City and Takao Goto at Chuo University for providing valuable comments and Taichi Mizunoya at the Tokyo Metropolitan Government for helping with the analysis. We also Tabhank for many useful comments from three autonomous referees.

**Conflicts of Interest:** The authors declare no conflict of interest.

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
