# Peer review of "Heterogeneity and Temporal Stability of Residential Water Use Responsiveness to Price Change"

_water, doi:10.3390/w13172329_

Round 1

Reviewer 1 Report

This paper would analyze the elasticities for residential water users. I think that this work is interesting and original because based on japaneese data.

But, I suggest :

  1. To check more water studies (Mayol & Staropoli 2021 ; Mayol 2017 ; Mayol & Porcher 2019 in France, which analyze the different demand responses to price-change ; Barraque et al.)
  2. To better explain the assumptions (rationality) suggested by the respons to the marginal-price. Following Mayol & Staropoli 2021 ; Mayol&Porcher 2019, and all the behavioral debates, you need to question more the economics hypotheses.

By including these comments, I think that this short paper would be published

Best,

Author Response

Please find an attached Word file.

Reviewer 2 Report

The submission "Heterogeneity and Temporal Stability of Residential Water Use Responsiveness to Price Change" presents an analysis of water consumption data from individual households in Hadano City, Kanagawa Prefecture, Japan. Longitudinal data over 5 years is available with 5 observations for each year representing bi-monthly observations. Only complete data is considered.

More care is needed to describe the data as well as the pre-processing, including elimination of certain observations, in order to indicate if and how any biases might have been introduced. The consideration of only 5 bi-monthly observations for each is not clear to me.

The data contains a tariff change after year two. This means that quite a strong increase in price after the first two years is visible in the data. The aim is now to assess how households react to these price changes, taking potential heterogeneity in reaction into account as well as assessing the stability for two time points. The quantity of interest is price elasticity. It would be good to clearly write down the formulas how this price elasticity is determined, i.e., which change in use as well as which change in price is considered to derive this quantity. When defining this quantity it would also be important to indicate how households might learn about the change and then adapt their behavior. It is unclear how the costs are determined and paid by the households. In other countries for example the last year consumption is usually used to predict the next year's consumption and use this together with the current price to determine monthly rates. Only afterwards, i.e., half a year after the year is complete, the actual consumption would be determined and in case of a reduction a refund granted. In such a setting it would be unclear how reactive one would expect households to be. In addition it is also unclear how expensive water is, i.e., what share of the household budget this would take. It could be that costs are negligible even after the increase. So more care is needed to motivate why a change of behavior is expected from the households.

The analysis is complicated because one wants to determine the causal effect of the change but the data contains only observations where the treatment was assigned. In fact one would want to compare the (unobserved hypothetical) water consumption without tariff change to the water consumption with tariff change in order to determine exactly the effect of the tariff change. More care is needed to determine the potential outcome without treatment and compare this to the observed outcome with treatment.

Separate analysis for odd-numbered and even-numbered months and in Figure 4 only odd-numbered months are plotted. However, it remains unclear why this distinction is made and why the data is not jointly analyzed. The authors also state that they find the congruence of results in Table 3 "interesting". One might, however, also claim that this should be expected for a well fitting model which captures the true data generating model which should be the same for both these data sets.

The heterogeneity analysis made needs to be explained in more detail. It is not clear which model is exactly fitted. It would be good to write this down. In addition it would be good to also include the AIC values in some table and complement them with the results for the BIC. The interpretation of the range of values for the class sizes only use the point estimates for the two different data sets. Usually one would rather base this on some confidence intervals for these parameter values and make the statements also according to the confidence level set. 

Overall the manuscript needs more care to clearly outline previous research and results and not refer vaguely to many, some or few; indicate the hypotheses and assumptions made regarding the effect and how the availability data allows to assess them including specific formulas; explain in more detail the data and the pre-processing as well as the analysis made and improve the visualizations. More descriptive statistics and visualizations might also help the reader to gain some understanding of the data. The current ones could certainly be improved to better convey information. 

Specific comments:

o The data is provided at https://doi.org/10.5061/dryad.pnvx0k6mk. The zip file contains two CSV files:

hadanogusu.csv
hadanokisu.csv

It is unclear what they represent as the analysis only refers to the analysis of Hadano City. More explanations need to be provided to link these files to the analysis and clarify this for the reader. 

o The formulations regarding elasticity are sometimes unclear because it is unclear if the absolute change is meant or really the sign taking into account when referring to lower or higher. It would be good to carefully read the manuscript regarding this aspect and formulate more carefully.

o It remains unclear why only Year 1 and Year 3 are considered to assess stability and Year 2 after the tariff change is not taken into account. It should also be clarified if a Welch t-test was used, given that the boxplots would indicate differences in variance.

o The scatter plots in Figures 1 and 3 might be improved by using the same plotting symbol e.g. a full bullet together with alpha shading to account for overplotting and / or by plotting kernel density estimates and / or loess regression lines to indicate association. It is rather awkward to use a reversed y-axis for Figure 3 as this might be rather unexpected for the reader and be perceived as confusing.

o Figure 2 should be interpreted with more care. This figure raises for example the question if Year 2 after the tariff change differs from the other two year as well as if the effect of the tariff change differs for the months. Overall this would then also indicate that a visualization using all monthly observations might be preferable to provide further insights into these issues.

o Table 3 seems incomplete. It would be good to add all estimated parameters to give a complete picture.

o When stating that no significant differences were detected one should also include the test statistics, degrees of freedom and p-values.

Author Response

Please find an attached Word file.

Reviewer 3 Report

This is a potentially great paper! As it stands though rather incomplete with few significant shortcomings. Please see attachment. Also notice that the "ratings" reflect the submitted version and not the actual potential which could easily excel in all rating categories.

Author Response

Please find an attached Word file.
